# Integrated effects of anaerobic soil disinfestation and beneficial microbes in strawberry production

Baker D. Aljawasim[1¤], Patricia Richardson[1], Chuansheng Mei[2], Robert L. Chretien[2], J. Scott Lowman[2], Jayesh B. Samtani[1]*

1 Hampton Roads Agricultural Research and Extension Center, School of Plant and Environmental Sciences, Virginia Tech, Virginia Beach, Virginia, United States of America, 2 The Institute for Advanced Learning and Research, Danville, Virginia, United States of America

¤ Current address: Department of Plant Protection, College of Agriculture, Al-Muthanna University, Samawah, Al Muthanna Province, Iraq

* jsamtani@vt.edu

## Abstract

Fruit rot diseases, including anthracnose fruit rot and Botrytis fruit rot caused by *Colletotrichum acutatum* and *Botrytis cinerea*, respectively, pose major challenges to sustainable strawberry production in the United States. Organic and small-scale growers require non-chemical alternatives to soil fumigation due to health, regulatory, or technical constraints. This study evaluated anaerobic soil disinfestation (ASD), beneficial bacteria (*Bacillus velezensis* IALR619 and TerraGrow-a commercial product), and their combinations as integrated strategies to manage fruit rot, suppress weeds, and enhance fruit quality. A split-plot field trial was conducted over two growing seasons (2022/23–2023/24) at the Hampton Roads AREC in Virginia Beach, VA. ASD significantly reduced AFR, BFR, and overall fruit rot incidence compared to non-fumigated and Pic-Clor 60-fumigated controls. Post-plant inoculation with *B. velezensis* IALR619, TerraGrow, or TerraGrow + Oxidate 5.0 further decreased AFR incidence. ASD also reduced weed density and improved postharvest fruit firmness, total soluble solids, and juice pH. *B. velezensis* IALR619 inoculation enhanced fruit firmness compared to uninoculated controls. The beneficial microbes with ASD offer a promising alternative to synthetic fumigation, reducing fungicide usage by organic growers, small farms, and resource-limited growers, while also improving strawberry fruit quality.

## 1. Introduction

Strawberry (*Fragaria ×ananassa* (Weston) Duchesne ex Rozier) is an important and valuable small fruit known for its distinctive taste and nutritional and health advantages, making it one of the most popular berries globally [1]. It also contains high amounts of vitamin C and folate, which have a positive effect on human health. Currently, strawberries are sold in the global market with an estimated value of USD

---

---

**Data availability statement:** All relevant data are within the manuscript.

**Funding:** This research was supported in part by the Northeast Sustainable Agriculture Research (NE-SARE) Program through the USDA Research and Education Grant (#LNE20-401) and by North American Strawberry Growers Association. The funders had no role in study design, data collection and analysis, decision to publish, or preparation of the manuscript.

**Competing interests:** The authors have declared that no competing interests exist.

20.22 billion in 2023, with projections indicating growth to USD 35.35 billion by 2032, and the United States accounts for 20% of global production [2]. However, consumer surveys and media reports indicate increasing dissatisfaction with strawberry flavor and inconsistent quality, as the majority of consumers prioritize an enjoyable taste experience [3]. Consequently, further research is needed to enhance strawberry fruit quality, particularly focusing on attributes such as firmness and total soluble solids, and fruit acidity especially in small farm and resource-limited growers. Small farms are generally characterized by limited acreage, lower annual sales, or small-scale production, while resource-limited growers are those constrained by inadequate access to capital, land, technology, infrastructure, and extension services, limiting their ability to invest in advanced production or adopt new technologies [4,5].

Strawberries are susceptible to a diverse range of plant pathogens, including anthracnose fruit rot (AFR) caused by *Colletotrichum acutatum* J.H. and Botrytis fruit rot (BFR) or grey mold caused by *Botrytis cinerea*, which are major fruit rot diseases of strawberries in the United States and other strawberry-growing regions worldwide [6,7]. Both these diseases can lead to substantial economic losses in the strawberry production due to their severe impacts [8].

Effective management of AFR and BFR is crucial to achieve the desired marketable yield. For many years, the primary method to prevent damage caused by fruit rot diseases, both preharvest and postharvest, has been the utilization of chemical-based control. However, growing interest has emerged in developing safe, sustainable, and cost-effective fumigation strategies for managing plant pathogens, prompted by the phase out of highly effective soil fumigants and the imposition of restrictions on other synthetic fumigants [9]. The need for alternative approaches has increased significantly among organic farmers and small-scale growers, who are unable to utilize synthetic fumigants. Furthermore, it is imperative to prioritize the development of alternative strategies for pathogen suppression to effectively mitigate the risks posed to both public health and the environment [10].

Another significant pest impacting strawberry production is weeds, especially those that grow in the planting hole or penetrate through plastic mulch, which can greatly influence strawberry yield and the quality of the fruit [11]. Weeds compete with strawberry plants for primary resources such as water, nutrients, and light, and can increase the risk of diseases, nematodes, and insect pests. Dense weed growth can impede harvesting by obscuring the fruits, increasing production costs due to the necessity for extensive weed control [12]. The most efficient and direct method for managing weeds is to fumigate the soil before planting or applying preplant herbicides. However, due to human health concerns, the need for untreated buffer areas within the farm, potential environmental harm, evolved chemical resistances, and the high cost of soil fumigation, many agricultural managers and growers are increasingly opting against their utilization [13,14]. Achieving such a challenging goal necessitates the implementation of more research to explore alternative non-fumigant strategies, not only to mitigate the negative impacts of weeds but also to enhance crop yield.

Anaerobic soil disinfestation (ASD) is a promising substitute to synthetic fumigation for managing diseases in strawberry production systems. Researchers in Japan and

the Netherlands initially documented this technique separately, which was subsequently modified for implementation in the United States to manage soil-borne pathogens in strawberry and vegetable fields [15,16]. The ASD procedure involves the introduction of an organic carbon source to the topsoil, subsequent irrigation to field capacity, and the application of a plastic film over the soil surface for a period of 2–5 weeks to maintain anaerobic conditions [17]. This sequence of actions was intended to minimize the rate of gas exchange and create anaerobic conditions in the soil. During this process, the soil undergoes a significant reduction, effectively eliminating harmful pathogens and pests present in the soil matrix [18]. Several pieces of evidence point towards the potential involvement of ASD in the development of disease-suppressive soils [19].

The impact of ASD on plant pests is not attributed to a single mechanism but rather results from the interplay of multiple processes working in concert. The initial mechanism pertains to fermentative decomposition, which under adequately anaerobic conditions involving carbon sources, results in a decrease in soil pH and redox potential (Eh). This process also facilitates the production of volatile organic compounds (VOCs) and volatile fatty acids (VFAs), ultimately contributing to the suppression of plant pathogens [20,21]. The efficacy of ASD in disease suppression has been attributed to alterations in soil microbial communities following treatment [22]. Certain soil microorganisms can actively suppress plant-pathogenic fungi by directly inhibiting their growth and competing for vital nutrients and the limited oxygen available in the soil [23]. Furthermore, the generation of $Fe^{2+}$ and $Mn^{2+}$ under low Eh conditions during ASD may exhibit disease-suppressing characteristics [24].

Beneficial microbes act as bio-control agents by inhibiting plant pathogens such as fungi, bacteria, and nematodes while simultaneously promoting plant growth. *Bacillus* spp. have emerged as promising candidates for effectively combating fruit rot disease in diverse hosts. *Bacillus* species are recognized for their capacity to generate antimicrobial substances and resilient endospores, enabling them to withstand environmental stresses while efficiently managing plant pathogens in both laboratory and field conditions [25]. Lipopeptides, in particular, are significant among these compounds, demonstrating broad-spectrum antimicrobial activity against bacteria, fungi, oomycetes, and viruses [26]. For example, anthracnose of tamarillo fruits (*Cyphomandra betacea*) caused by *C. acutatum* was completely suppressed by the cell free supernatant (CFS), iturin A, and fengycin C, while *Bacillus subtilis* cells reduced symptoms by 76%; similarly, grey mold in chrysanthemum flowers (*Chrysanthemum morifolium* Ramat) caused by *B. cinerea* was inhibited by 72% with lipopeptides and by 39% with *B. subtilis* EA-CB0015 cells [27,28]. Under greenhouse conditions, *Bacillus velezensis* strains suppressed the growth of the strawberry pathogen (*C. gloeosporioides*), and led to increased marketable fruit yield at some grower locations in an open-field environment [29]. Five specific strains from the *Bacillus* genus demonstrated the ability to suppress over 80% of the mycelial growth of *B. cinerea* through the release of diffusible compounds, and they achieved an impressive 90% inhibition by producing volatile antifungal compounds [30]. The mechanisms through which they operate are diverse and encompass competition for space and/or nutrients, parasitic interactions, and the synthesis of harmful metabolites [31]. Thus, the opportunities provided by the application of organic substances, biofertilizers, or biostimulants as an element of the ASD technique to manage fruit rot diseases and increase the growth and productivity of strawberries are highly promising for supporting sustainable crop production. The main goals of this study were to evaluate the impact of ASD, beneficial microbes, and their combination on managing diseases such as fruit rot, including AFR and BFR, controlling weeds, and enhancing crop yield and postharvest fruit quality. To our knowledge, this is the first field study integrating ASD with beneficial microbes for annual strawberry crop production.

## 2. Materials and methods

### 2.1 Study site

A study was established at the Hampton Road Agricultural Research and Extension Center, Virginia Beach, Virginia (36 °9′ N, 76 °2′ W; 3.7m elevation) during the 2022/23 and 2023/24 growing seasons with four blocks or replicates. The soil

at the experimental site was a tetotum loam with a 0–2% slope, characterized as sandy loam with a deep profile and moderate drainage, originating from loamy fluvial and marine sediments. The study relied on the natural occurrence of fruit rot diseases in the field. Throughout both growing seasons, the older section of the site had a documented history of strawberry cultivation. The study involved two blocks located within this older section and two blocks situated in the new section. In contrast to the cultivated areas, the new section primarily consisted of grassy vegetation maintained by regular mowing. Soil tests were conducted before planting in both seasons, and soil samples from both sections of the site were sent to the Virginia Tech Soil Testing Laboratory in late July. The soil test report included recommendations for adjusting the pH of the new section during both growing seasons. The soil pH of the new section was adjusted to 6.2 by applying limestone at rates of 3,362 kg ha$^{-1}$ and 5,043 kg ha$^{-1}$, respectively, in the two seasons. The ground was then tilled to a depth of 15 cm below the surface in the designated research area during field preparation.

## 2.2 Experimental design

The study was conducted as a split–split plot randomized complete block design to efficiently evaluate multiple factors under field conditions, particularly where one factor necessitated larger experimental units than others, with four replications conducted under annual hill plasticulture production in both seasons. 'Chandler' is a common cultivar used for annual hill plasticulture cultivation in Virginia and the South Atlantic region of the U.S. [32]. Preplant soil treatments (ASD, fumigation, and non-fumigated control) were assigned to the main plots using a randomized complete block design (RCBD), with post-plant beneficial microbe treatments randomly allocated to the subplots (Table 1). The main plot dimensions measured 14.5 m in length, 0.7 m wide on the bed tops, and 0.15 m high. The orientation of the beds was north-south. Before bed formation, synthetic nitrogen fertilizers (urea, 42-0-0, PCS Sales, Inc., Northbrook, IL, USA) were applied at a rate of 117 kg ha$^{-1}$ to non-fumigated plots and plots treated with Pic-Clor 60 [1,3-dichloropropene plus chloropicrin (40:60, w/w)] at 196 kg ha$^{-1}$. For ASD plots, brewer's spent grain (BSG) from Commonwealth Brewing Company in Virginia Beach, VA, USA, was used as the carbon source without synthetic nitrogen fertilizers. Fresh BSG typically has a

**Table 1. List of the preplant and post-plant treatments during the 2022/23 and 2023/24 strawberry growing seasons.**

| Preplant Treatments | Post-plant Treatments | Abbreviation | Application rate |
|---|---|---|---|
| UTC[a] | N[d] | N | |
| UTC | *B. velezensis* IALR619 | C-BV | $OD_{600} = 1.0$ |
| UTC | TerraGrow[e] | C-TG | 1.68 kg ha$^{-1}$ |
| UTC | Oxidate 5.0[f] +TerraGrow | C-OTG | 23,384 L ha$^{-1}$ + 1.68 kg ha$^{-1}$ |
| Pic-Clor 60[b] | N | PIC | 196 kg ha$^{-1}$ |
| Pic-Clor 60 | *B. velezensis* IALR619 | PIC-BV | $OD_{600} = 1.0$ |
| Pic-Clor 60 | TerraGrow | PIC-TG | 1.68 kg ha$^{-1}$ |
| Pic-Clor 60 | Oxidate 5.0 + TerraGrow | PIC-OTG | 23,384 L ha$^{-1}$ + 1.68 kg ha$^{-1}$ |
| ASD[c] | N | ASD | |
| ASD | *B. velezensis* IALR619 | ASD-BV | $OD_{600} = 1.0$ |
| ASD | TerraGrow | ASD-TG | 1.68 kg ha$^{-1}$ |
| ASD | Oxidate 5.0 + TerraGrow | ASD-OTG | 23,384 L ha$^{-1}$ + 1.68 kg ha$^{-1}$ |

[a] UTC: Non-fumigated treatments.

[b] Pic-Clor 60: [1,3-dichloropropene plus chloropicrin (40:60, w/w)] shank fumigated at 196 kg ha$^{-1}$.

[c] ASD: anaerobic soil disinfestation using brewer's spent grain (4.5 tons/ ha$^{-1}$) as nitrogen source and paper mulch (4.8 tons/ ha$^{-1}$) as a carbon source.

[d] N: non-inoculated treatments.

[e] TerraGrow: a five-strain formula of *Bacillus* spp., Biosafe systems, CT, USA.

[f] Oxidate 5.0: 27% hydrogen peroxide and 5% peroxyacetic acid, Biosafe systems, CT, USA.

carbon-to-nitrogen ratio of approximately 14:1, with a composition that includes about 44.8% total carbon, 3.8% total nitrogen, 15–24% protein, 17–25% cellulose, 20–30% hemicellulose, and 12–27% lignin (all on a dry weight basis), making it a potentially valuable carbon source that could reduce the need for additional nitrogen inputs [33]. The application rates were determined according to the recommendation of previous studies [34]. Specifically, the carbon application rate was set at 4 mg C/g soil, considering a soil depth of 15 cm and a soil density of 1.08 g/cm. The selection of carbon sources was also affected by their availability, our previous study, and cost in the region during the preparation of the strawberry beds. These amendments, including carbon sources and fertilizers, were manually broadcast before the final bed preparation on September 1, 2022, and September 8, 2023. They were incorporated to a depth of 15 cm using a single-pass machine that cultivated shaped beds, installed drip tapes, and laid a 1.25 mil virtually impermeable plastic film (VIF, Raven Industries Engineered Films Division, Sioux Falls, SD, USA) over all beds, including controls. Irrigation and fertigation throughout the growing season were managed using a 0.38 mm drip line with a 30.5 cm emitter spacing and a flow rate of 1.7 L per minute (Berry Hill Irrigation, Inc., Buffalo Junction, VA 24529), placed approximately 5 cm below the bed surface to ensure soil saturation and nutrient delivery.

The ASD treatments were initiated on September 8, 2022, and September 12, 2023, with the initial step being the irrigation of the beds to uphold soil moisture at a field capacity of 23% volumetric water content (VWC). Before the implementation of the ASD, the soil moisture levels were approximately 12% VWC and 14% VWC, respectively. Soil moisture levels were measured using a Field Scout TDR 100 soil moisture sensor (Spectrum Technologies, Inc., Aurora, IL, USA), which was periodically employed to determine the optimal timing and duration of irrigation sessions throughout the 21 days of ASD period. The fumigant treatment, Pic-Clor 60 [1,3-dichloropropene plus chloropicrin (40:60, w/w)], shank fumigated at 196 kg ha$^{-1}$ on September 2, 2022, and September 12, 2023. Before punching holes for the transplants, Italian ryegrass (Lolium multiflorum Lam.) was broadcast seeded on October 10, 2022, and October 2, 2023, at 44.8 kg ha$^{-1}$ within the furrows to improve drainage, establish a grass walkway, and reduce establishment of weeds.

Following the application of preplant treatments, each main plot was further divided into four subplots, each 2.4 m long with a buffer of 1.5 m between plots, resulting in a total of forty-eight plots (Table 1). On October 17, 2022, and October 12, 2023, fourteen strawberry plug plants of the 'Chandler' variety were planted in staggered double rows. There was a spacing of 30 cm between the rows and 36 cm between each plant in each plot. Unless otherwise specified, standard post-plant fertilization and irrigation practices for conventional strawberries in the area were followed during both growing seasons. For post-plant treatments, strawberry plants were treated with B. velezensis IALR619, TerraGrow, and a combination of Oxidate® 5.0 and TerraGrow, applied as soil drenches directly into the planting hole. B. velezensis IALR619 was obtained from the Institute for Advanced Learning and Research Laboratory in Danville, VA, diluted to 24.4 ml plant$^{-1}$, and applied twice in the fall and twice in the spring during each of two consecutive growing seasons. In contrast, TerraGrow and the combination of Oxidate® 5.0 and TerraGrow were each applied three times during the fall and three times during the spring in both growing seasons. For the combination treatment of Oxidate® 5.0 and TerraGrow, Oxidate® 5.0 was applied first, followed by a four-hour interval before the subsequent application of TerraGrow. No fungicides were applied throughout both growing seasons, as the focus was solely on the treatments implemented within the strawberry production system.

### 2.3 Sensor installation

On September 8, 2022, and September 12, 2023, redox potential sensors (ORP2000 Extended Life ORP Sensor, Sensorex, Garden Grove, CA, USA) were placed at a depth of 15 cm in the center of the bed to assess soil anaerobic conditions during the ASD period. These sensors were linked to an automated data-logging system (CR-1000, Campbell Scientific, Logan, UT, USA). At the same depth as the ORP sensors, soil temperature sensors (U12 Deep Ocean Temperature Data Logger, Onset, Bourne, MA, USA) were also installed. Except the Pic-Clor 60 treatment, each treatment included two plots equipped with sensors, and both types of sensors recorded data every 60 minutes throughout the treatment period.

## 2.4 Data collection

Strawberry fruits were harvested twice weekly from March 31, 2023, to June 20, 2023, and from April 16, 2024, to June 13, 2024. During each harvest session, the fruits were sorted into marketable and non-marketable categories according to their size (<10 g), diseases, deformities, or ripeness. Deformed fruits were described as berries exhibiting physical characteristics that rendered them unmarketable, such as nubbins or button berries. Fruits infected by AFR and BFR were weighed separately. Fruits were categorized as diseased when visible lesions were present, and as healthy when no symptoms were observable. In cases where there was uncertainty about the pathogen causing the lesion, the diagnosis was verified by examining conidia using a dissecting microscope. The biomass of the marketable and non-marketable fruits was summed to determine the total yield. Marketable and total yield per plant were calculated by summing up the cumulative berry yield data from each replicate over all harvests and dividing that amount by the number of plants in each replicate.

For post-harvest measurements, five marketable fruits were sampled randomly for each treatment in May and June to assess the fruit firmness, total soluble solid content, and pH of the fruit juice. Fruit firmness was determined using a tabletop fruit texture analyzer (GS-15 Fruit Texture Analyzer; QA Supplies, Norfolk, VA, USA), with five marketable fruits per replicate tested at every other harvest date. The berries were then processed by first removing the calyxes, then crushing them, and finally sieving to filter the juice from the pulp. Total soluble solids (°Brix) were tested using a refractometer (MA871 Refractometer, Milwaukee, Rocky Mount, NC) on the same batch of fruits, and pH was assessed using a pH tester electrode (combo pH/conductivity/TDS tester, HANNA Instruments, Smithfield, RI, USA).

Weeds were counted by species at each crop plant hole, and the total weed density was determined by adding the counts of all species within each replicate. Weed assessment dates for both growing seasons were selected based on when the emerging weeds began to compete with the crop in the planting hole. Weed data were collected on December 21, 2022, in the first season, and on December 18, 2023, and March 19, 2024, in the second season. Monthly plant vigor ratings were recorded for each plot from November to June and averaged across the growing season. The vigor of all plants within each replicate was evaluated on a scale from 0 (complete plant mortality) to 10 (exceptionally vigorous plants). Plant canopy diameter measurements (width × length) were conducted independently for each replicate on December 8, 2022, and April 21, 2023, during the 2022/23 growing season. For the 2023/24 growing season, canopy measurements were obtained on December 7, 2023, and April 22, 2024.

## 2.5 Statistical analyses

Initially, the data were evaluated for normality and the consistency of variance. If these assumptions were not satisfied, appropriate transformations were applied. The data that satisfied the normality assumptions were subjected to analysis of variance (ANOVA) with growing season, main treatments (preplant treatments), sub-treatments (post-plant treatments), and their interaction as fixed effects. Fisher's protected least significant difference (LSD) test was then performed using JMP® Pro 16.0.0 (SAS Institute Inc., Cary, NC, USA). The average temperature during the 21-day treatment period was calculated, and the cumulative redox potential (CuEh) was determined using hourly average redox potential values. For each hour, the deviation of the hourly mean redox potential from the critical redox potential (CEh) was calculated, and these deviations were summed over the three-week ASD treatment period. The threshold redox potential (CEh) was calculated using the equation CEh = 595 mV − 60 mV × soil pH [35], where values below this threshold indicate anaerobic soil conditions. The intensity and duration of anaerobic conditions were further quantified over the three weeks using the cumulative Eh (mV·h) below 200 mV as an indicator [36]. The data on fruit biomass affected by AFR and BFR were log (x + 1) transformed to meet normality assumptions. Consequently, the original least-squares means are displayed in the figure, whereas the separation letters are derived from the transformed mean values. To satisfy normality assumptions, the weed density data for all species were subjected to log (x + 1) transformation. The results presented in the table are the least-squares means, and the separation letters are based on transformed means.

## 3. Results and discussion

### 3.1 Cumulative soil anaerobic conditions and temperature

The redox potential and soil temperature data were collected from plots treated with ASD and non-fumigated treatments. The development of anaerobic conditions exhibited a highly significant difference (P = 0.001) between ASD and non-fumigated control treatments. The ASD treatments demonstrated strong anaerobic conditions (Eh 200 to −400 mV) during the 2023/24 growing season, relative to the 2022/23 growing season, which ranged from 200 to 100 mV (Fig 1). The redox potential (Eh) readings of ASD observed in this study were in the range of 200 to −400 mV during both growing seasons, which were similar to those recorded in other studies [36,37]. These low redox values indicate the establishment of strictly anaerobic conditions, which can help decrease pathogen populations. However, these anaerobic conditions may also harm beneficial soil microorganisms, particularly those that play a role in nutrient cycling and symbiotic relationships with plants [17].

The analysis of soil temperature revealed no significant differences between the ASD and untreated control treatments in terms of mean and maximum temperatures at a depth of 15 cm over the two growing seasons, which ranged from 20 to 40.7 °C for the 2022/23 growing season and 19–35°C for the 2023/24 growing season. Nonetheless, both the ASD and non-fumigated control treatments yielded significantly higher temperatures than the ambient air temperature recorded during the treatment period (P = 0.001; S1 Fig). A similar difference in temperatures between soil and air was documented in our earlier greenhouse trials [38]. Although ASD does not need high temperatures to work effectively, the soil temperature, along with the carbon source, is essential in influencing how well ASD treatment controls diseases [36]. For example, another study indicated that at elevated soil temperatures, the reduction of pathogens due to the ASD effect was approximately 10% greater compared to moderate and lower temperatures [23].

### 3.2 Fruit rot diseases

The interaction between the preplant treatments and the growing season, along with the post-plant treatments, did not show any statistically significant effect on AFR fruit biomass; therefore, only the main effect data is presented. Although a statistically significant interaction was observed between the growing season and post-plant treatments, the independent effect of post-plant treatments was not statistically significant in either growing season. The ASD treatment had significantly less AFR (0.97 g plant$^{-1}$ by biomass, P = 0.003) compared to the Pic-Clor 60 and non-fumigated control treatments

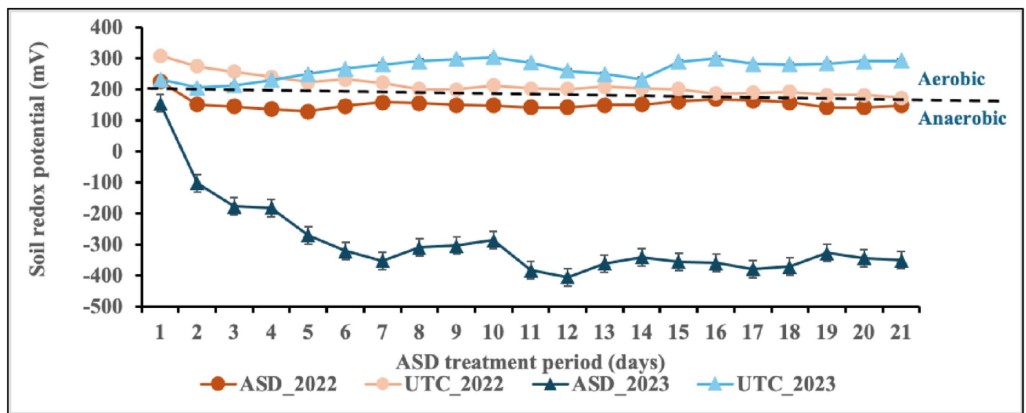

**Fig 1. The mean daily soil redox potential (Eh) was influenced by anaerobic soil disinfestation (ASD) during the three-week treatment period.** Value is the mean of n = 4 replicates for each treatment during for two growing seasons 2022/23 and 2023/24. The dotted line indicates the critical redox potential value of 200 mV. ASD_2022: ASD with brewer's spent grain in the 2022/23 growing season. UTC_2022: Untreated control during the 2022/23 growing season. ASD_2023: ASD during the 2023/24 growing season. UTC_2023: non-fumigated control during the 2023/24 growing season.

(4.25 and 6.62 g plant$^{-1}$, respectively) across both growing seasons (Fig 2). The interaction between the preplant treatments and the growing season was significant for BFR (P = 0.002) and for other diseases (P=0.001). Hence, data for the two growing seasons are presented separately. ASD treatment significantly reduced the biomass of infected fruit caused by BFR or other diseases when compared to both the non-fumigated control and the Pic-Clor 60 treatment across both growing seasons (Figs 3 and 4). The suppression of these diseases may be attributed to the anaerobic decomposition of the carbon source in ASD, which is typically associated with the release of volatile compounds and organic acids, reduction in soil pH, and alterations in the soil microbial community [21,39]. In the 2022/23 growing season, the application of post-plant treatments *B. velezensis* IALR619, TerraGrow, and Oxidate 0.5 plus TerraGrow significantly reduced AFR fruit weight to 3.05, 3.24, and 4.43 g plant$^{-1}$, respectively, compared with 16.1 g plant$^{-1}$in the non-inoculated control (Fig 2). These results align with previous studies showing that certain *Bacillus* species possess antifungal properties against *C. acutatum* and *C. gloeosporioides* by producing substantial amounts of chitinase, protease, and cellulase enzymes, which are recognized for targeting the fungal cell wall, causing its breakdown and ultimately leading to the lysis of the fungal cell [40,41]. This study, consistent with other research, demonstrated that *B. velezensis* TS3B-45 exhibited strong antifungal activity, inhibiting fungal pathogens by 80% in vitro and effectively controlling various plant pathogens, including *Colletotrichum* spp. and other fungal plant pathogens [42]. In a previous study, researchers detected some lipopeptides, such as surfactin, and iturin in cultures of the *B. velezensis* IALR619 strain, the same isolate used in our study, which may contribute to disease suppression [29]. Other studies have indicated that *Bacillus* spp. can effectively suppress anthracnose rot caused by *C. acutatum* in postharvest loquat fruit (*Eriobotrya japonica* Lindl.), potentially by directly inhibiting pathogen growth and indirectly enhancing disease resistance in the host through the activation of two defense-related enzymes, chitinase and β-1,3-glucanase [43]. *B. velezensis* IALR619, TerraGrow, and Oxidate 0.5 + TerraGrow, did not show any effect on BFR control (Fig 3). During the 2023/24 growing seasons, the incidence of all reported fruit rot diseases was low, and we did not observe any effect from microbial inoculation. Conversely, the effectiveness of post-plant inoculation is greatly affected by environmental conditions, which play a crucial role in its ability to effectively suppress diseases. Factors such as temperature, humidity, and soil characteristics can either improve or impede the performance of these *Bacillus* species, as these conditions influence the survival, growth, and activity of the agents [44].

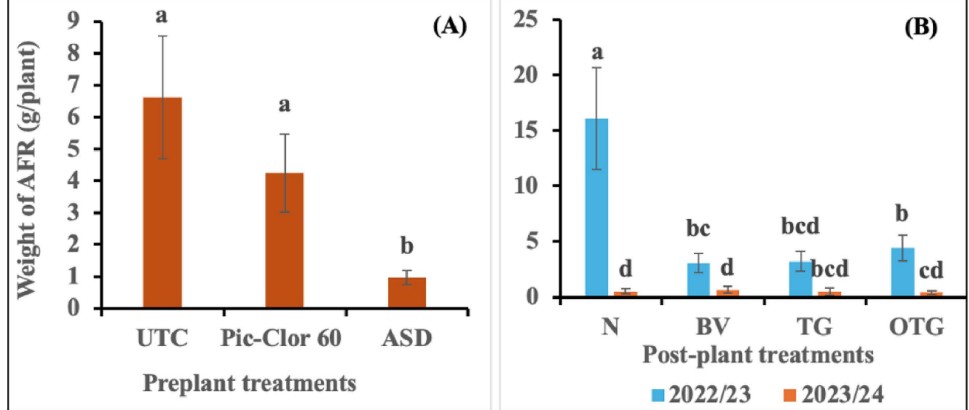

**Fig 2. Effect of preplant treatments (a) and post-plant treatments (b) on anthracnose fruit rot (AFR) fruit biomass during the 2022/23 and 2023/24 growing seasons.** UTC: non-fumigated control; Pic-Clor 60: [1,3-dichloropropene plus chloropicrin (40:60, w/w)] shank fumigated at 196 kg ha$^{-1}$; ASD: anaerobic soil disinfestation using brewer's spent grain (4.5 tons/ ha$^{-1}$) as nitrogen source and paper mulch (4.8 tons/ ha$^{-1}$) as a carbon source. The post-plant treatments: N: non-inoculated control; BV: inoculated with *B. velezensis* IALR619, OD$_{600}$ = 1.0; TG: inoculated with TerraGrow; OTG: inculcated with Oxidate 5.0 + TerraGrow. The means in the bar graphs with the same letters indicate no statistically significant differences (*P* < 0.05) using Fisher's Least Significant Difference (LSD). Error bars indicate the standard error of the mean.

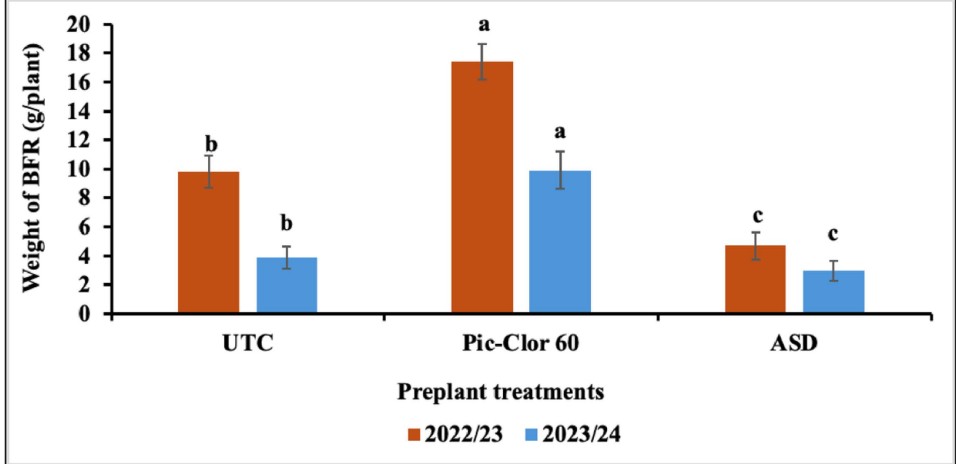

**Fig 3. Effect of preplant treatments on botrytis fruit rot (BFR) diseases during the 2022/23 and 2023/24 growing seasons.** UTC: non-fumigated control; Pic-Clor 60: [1,3-dichloropropene plus chloropicrin (40:60, w/w)] shank fumigated at 196 kg ha⁻¹; ASD: anaerobic soil disinfestation using brewer's spent grain (4.5 tons/ ha⁻¹) as nitrogen source and paper mulch (4.8 tons/ ha⁻¹) as a carbon source. The means with the same letters indicate no statistically significant differences ($P<0.05$) using Fisher's Least Significant Difference (LSD). Error bars indicate standard error of the mean.

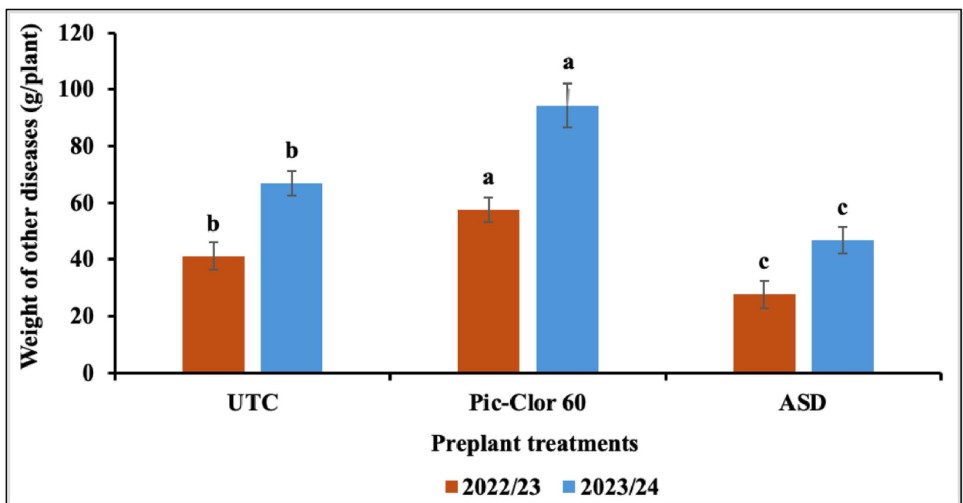

**Fig 4. Effect of preplant treatments on the biomass of infected fruits by other diseases during the 2022/23 and 2023/24 growing seasons.** UTC: non-fumigated control; Pic-Clor 60: [1,3-dichloropropene plus chloropicrin (40:60, w/w)] shank fumigated at 196 kg ha⁻¹; ASD: anaerobic soil disinfestation using brewer's spent grain (4.5 tons/ ha⁻¹) as nitrogen source and paper mulch (4.8 tons/ ha⁻¹) as a carbon source. Means are represented by bars with the same letters, indicating no statistically significant differences ($P<0.05$) using Fisher's Least Significant Difference (LSD). Error bars indicate the standard error of the mean.

### 3.3 Weed density

Common vetch (*Vicia sativa* L.), cudweed (*Gnaphalium* spp.), henbit (*Lamium amplexicaule* L.), ryegrass (*Lolium perenne* L.), and white clover (*Trifolium repens* L.) were the dominant weed species in both growing seasons. A significant interaction between growing season and preplant treatment was observed for henbit, ryegrass, white clover, cumulative weed count, and total fresh biomass (Table 2). ASD lowered the common vetch density (0.65 plants per planting hole) compared

Table 2. Weed counts in the plant hole for the 2022/2023 and 2023/2024 growing seasons influenced by preplant treatments in annual hill plasticulture of strawberry production.

| Treatment | Common vetch | | Cudweed | | Henbit | | | | Ryegrass | | | | White Clover | | | |
|---|---|---|---|---|---|---|---|---|---|---|---|---|---|---|---|---|
| | | | | | 2022/23 | | 2023/24 | | 2022/23 | | 2023/24 | | 2022/23 | | 2023/24 | |
| --------------------------- (Weed density/planting hole) --------------------------- | | | | | | | | | | | | | | | | |
| UTC[a] | 2.13 | a[d] | 1.22 | a | 1.06 | a | 6 | a | 1.56 | a | 2.31 | a | 5.25 | a | 8 | a |
| Pic-Clor 60[b] | 1.10 | ab | 0.28 | b | 0 | b | 0.06 | b | 0.81 | b | 0 | b | 2.81 | b | 0.56 | b |
| ASD[c] | 0.65 | b | 1.09 | a | 0.06 | b | 1.5 | b | 0.81 | b | 1.75 | a | 3.06 | b | 5.56 | a |
| Pr>F[e] | 0.0246 | | 0.0041 | | <0.0001 | | | | 0.0004 | | | | <0.0001 | | | |

[a] UTC: Non-fumigated treatment.

[b] Pic-Clor 60 [1,3-dichloropropene plus chloropicrin (40:60, w/w)] shank fumigated at 196 kg ha$^{-1}$.

[c] ASD: anaerobic soil disinfestation using brewer's spent grain (4.5 tons/ ha$^{-1}$) as a nitrogen source and paper mulch (4.8 tons/ ha$^{-1}$) as a carbon source.

[d] Means with identical letters within a column do not differ significantly when analyzed using the least significant difference test at P ≤ 0.05.

[e] A significant interaction between the growing season and treatment was observed using ANOVA for henbit, ryegrass and white clover. For common vetch and cudweed, only the main effect of treatment was significant, resulting in the data being combined across the two growing seasons.

to the non-fumigated control. Nevertheless, the density of cudweed in the ASD treatment did not differ significantly from that in the non-fumigated control, and a lower density was observed in the Pic-Clor 60 treatment. For both growing seasons, henbit density in the ASD and Pic-Clor 60 treatment was significantly lower (P = 0.0001) than that in the non-fumigated control. During the 2022/23 growing season, ryegrass and white clover densities were lower (P = 0.0004 and P = 0.0001, respectively) in the ASD and Pic-Clor 60 treatment than in the non-fumigated control. In the 2023/24 growing season, no significant difference in ryegrass and white clover density was observed between the ASD plots and the non-fumigated control. These results may be due to the introduction of these weeds into the field with strawberry transplants or during the sowing of ryegrass seeds in the furrows, which could lead to their accumulation in planting holes after the termination of ASD. An additional challenge is that white clover shows tolerance to various herbicides typically used for controlling broadleaf weeds, making its management more difficult [45]. ASD plots exhibited significantly lower cumulative weed counts (P = 0.0001) and total fresh biomass (P = 0.0001) in both seasons compared with the non-fumigated control. During the 2022/23 growing season, the performance of ASD was similar to that of Pic-Clor 60. In the 2023/24 growing seasons, the lowest cumulative weed count and fresh biomass were recorded in the Pic-Clor 60 treatment (Table 3).

Table 3. Cumulative weed counts and weed biomass in the plant hole for the 2022/2023 and 2023/2024 growing seasons influenced by preplant treatments in annual hill plasticulture strawberry production.

| Treatment | Cumulative weed count | | | | Fresh biomass (g) | | | |
|---|---|---|---|---|---|---|---|---|
| | 2022/23 | | 2023/24 | | 2022/23 | | 2023/24 | |
| -------------------------- (Weed density/planting hole) ------------------------ | | | | | | | | |
| UTC[a] | 13.1 | a[d] | 33.9 | a | 2.37 | a | 57.20 | a |
| Pic-Clor 60[b] | 5.75 | b | 2.56 | c | 0.95 | b | 9.42 | c |
| ASD[c] | 5.69 | b | 16.6 | b | 0.83 | b | 20.2 | b |
| Pr>F[e] | <0.0001 | | | | <0.0001 | | | |

[a] UTC: Non-fumigated treatment.

[b] Pic-Clor 60 [1,3-dichloropropene plus chloropicrin (40:60, w/w)] shank fumigated at 196 kg ha$^{-1}$.

[c] ASD: anaerobic soil disinfestation using brewer's spent grain (4.5 tons/ ha$^{-1}$) as a nitrogen source and paper mulch (4.8 tons/ ha$^{-1}$) as a carbon source.

[d] Means with identical letters within a column do not differ significantly when analyzed using the least significant difference test at P ≤ 0.05.

[e] A significant interaction between the growing season and treatment was observed using ANOVA.

While previous studies have reported inconsistent weed control using ASD [23,46], this study showed that although the effectiveness of ASD differed among weed species, it consistently managed to suppress several types of weeds across both growing seasons. Our findings are consistent with those from Liu et al. [47,48], who demonstrated that the germination of common chickweed (*Stellaria media* L.), redroot pigweed (*Amaranthus retroflexus* L.), and white clover (*Trifolium repens* L.) seeds was similarly inhibited by ASD application in greenhouse settings. While the exact cause of ASD remains unclear, various factors could play a role in the reduction of weed density, including hypoxic conditions for imbibed seeds during the ASD treatment period, which may result in the loss of viability of imbibed seeds. The disruption of soil caused by the introduction of carbon sources and subsequent watering to promote the decomposition of these sources may lead to the establishment of stale seedbed conditions at the onset of ASD [49]. Furthermore, the buildup of metabolic byproducts, primarily organic fatty acids, along with the reduction of soil due to oxygen depletion, also results in alterations to soil pH during the ASD treatment period [50]. Later research has identified short-chain fatty acids, including groups of acids, as possible bioactive compounds released during the anaerobic decomposition of organic contents in soil, potentially contributing to pest and weed control [51–53]. In general, the post-plant treatments with the inoculation with beneficial microbes, including *B. velezensis* IALR619, TerraGrow, and a combination of Oxidate 0.5 with TerraGrow, did not significantly affect the weed density for all weed species during both growing seasons. The application of post-plant treatments was ineffective for weed control, likely due to the timing of application, as treatments were applied two weeks after planting, when weeds may have already established in the planting holes. This limitation underscores the necessity of aligning biocontrol strategies with critical growth stages of both target weeds and crops to enhance efficacy, as previous research has emphasized the importance of application timing in the successful implementation of biocontrol agents [54].

### 3.4 Strawberry fruit yield

The interaction between the growing season and preplant treatment was not significant for cumulative marketable yield, total yield, small fruit, or deformed fruit. However, the main effects are significantly different. In addition, there was a significant interaction between the preplant treatments and the growing season for non-marketable yield; therefore, the data are presented for two growing seasons (Table 4). The ASD treatment resulted in the lowest cumulative marketable yield and total yield among the pre-plant treatments (11,401 and 18,347 g plant$^{-1}$, respectively), whereas the Pic-Clor 60 treatment produced the highest yields (16,533 and 26,035 g plant$^{-1}$, respectively). This study showed that while cumulative Eh values were consistently lower in ASD treatments with different carbon sources compared to untreated plots, the extent of Eh reduction did not directly translate into improvements in strawberry yield. Our findings are inconsistent with other studies that showed ASD did not increase the yield for either tomatoes or bell peppers [55]. ASD treatments were consistently linked to improved or unaffected strawberry yields in different studies, and these improved yields were linked to the management of certain soil-borne pathogens [36,56,57]. In this study, the use of locally sourced carbon materials, such as BSG, in ASD treatments may have resulted in reduced levels of plant-available nitrogen in the early stages of plant growth, potentially lowering strawberry yield. In addition, ammonium ($NH_4^+$) content increased, and urease activity went remarkably high in the soil during previous ASD studies [58]. The nitrification process was primarily inhibited by the low oxygen concentration in the soil during the anaerobic process, which subsequently had a detrimental effect on plant growth in the early developmental stages due to reduced nitrogen uptake [58]. The application of synthetic preplant fertilizers in non-fumigated and fumigated plots supplied an easily accessible nitrogen source, promoting higher strawberry yields. Therefore, this challenge can be mitigated by enriching ASD-treated soils with supplementary synthetic nitrogen to counteract nitrogen depletion and ensure adequate nutrient availability for strawberry plants, particularly during the early stages of growth.

A significant reduction in non-marketable yield was observed in the ASD treatment (194 g plant$^{-1}$) across both growing seasons, compared with the non-fumigated control (231 g plant$^{-1}$) and the Pic-Clor 60 treatment (292 g plant$^{-1}$). The Pic-Clor 60 treatment resulted in the lowest amount of small fruit among all the preplant treatments. The ratio of marketable

Table 4. Effect of preplant treatments on strawberry yield in 2022/2023 and 2023/2024 growing seasons on annual plasticulture strawberry production.

| Treatment | Marketable yield | Nonmarketable yield | | Total yield | Ratio of Nonmarketable Yield (%) | Ratio of Marketable Yield (%) |
|---|---|---|---|---|---|---|
| | | 2022/23 | 2023/24 | | | |
| | ---------------------------- g plant $^{-1}$ --------------------------------------------- | | | | | |
| UTC[a] | 480 b[d] | 194 b | 268 b | 711 b | 33.3 | 66.7 |
| Pic-Clor 60[b] | 576 a | 225 a | 359 a | 868 a | 34.2 | 65.8 |
| ASD[c] | 368 c | 163 c | 226 c | 562 c | 35.1 | 64.7 |
| Pr>F Treatment[e] | <0.0001 | <0.0001 | | <0.001 | 0.25002 | 0.25002 |
| P season effect[f] | 0.99246 | <0.0001 | | 0.0042 | <0.0001 | <0.0001 |
| P t x season[g] | 0.51278 | 0.01103 | | 0.4510 | 0.21305 | 0.21305 |

[a] UTC: Non-fumigated treatments.

[b] Pic-Clor 60 [1,3-dichloropropene plus chloropicrin (40:60, w/w)] shank fumigated at 196 kg ha$^{-1}$.

[c] ASD: anaerobic soil disinfestation using brewer's spent grain (4.5 tons/ ha$^{-1}$) as a nitrogen source and paper mulch (4.8 tons/ ha$^{-1}$) as a carbon source.

[d] Means with identical letters within a column do not differ significantly when analyzed using the least significant difference test at P ≤ 0.05.

[e] P-value is of the preplant treatment main effect. A significant interaction between growing season and preplant treatment was observed using Fisher's Least Significant Difference (LSD) for non-marketable yield. For marketable yield and total yield, only the preplant treatment main effect was significant, resulting in the data being combined across the two growing seasons.

[f] P-value presented is the main effect of growing season.

[g] P-value represents the interaction between the preplant treatment and growing season.

and non-marketable yields to total yield was not statistically significant for any of the treatments. During both growing seasons, the ASD treatment significantly decreased the weight of deformed or damaged fruits (Table 5). Multiple researchers have reported that ASD can induce significant alterations in the soil microbiome, potentially leading to the establishment

Table 5. Effect of preplant treatments on strawberry fruit yield parameters in 2022/2023 and 2023/2024 growing seasons on annual plasticulture strawberry production.

| Treatment | Small fruit | | Deformed fruit | | Damaged fruit | | | |
|---|---|---|---|---|---|---|---|---|
| | | | | | 2022/23 | | 2023/24 | |
| | ------------------------ g plant $^{-1}$ ------------------------------------- | | | | | | | |
| UTC[a] | 95.4 | a[d] | 40.2 | a | 17 | b | 41 | b |
| Pic-Clor 60[b] | 84.1 | b | 48 | a | 27.8 | a | 104 | a |
| ASD[c] | 95.8 | a | 34.5 | b | 14.8 | c | 29.2 | c |
| Pr>F Treatment[e] | 0.0217 | | 0.0004 | | <0.0001 | | | |
| P season effect[f] | <0.0001 | | <0.001 | | <0.0001 | | | |
| P t x season[g] | 0.6321 | | 0.0873 | | 0.03807 | | | |

[a] UTC: Non-fumigated treatments.

[b] Pic-Clor 60 [1,3-dichloropropene plus chloropicrin (40:60, w/w)] shank fumigated at 196 kg ha$^{-1}$.

[c] ASD: anaerobic soil disinfestation using brewer's spent grain (4.5 tons/ ha$^{-1}$) as a nitrogen source and paper mulch (4.8 tons/ ha$^{-1}$) as a carbon source.

[d] Means with identical letters within a column do not differ significantly when analyzed using the least significant difference test at P ≤ 0.05.

[e] P-value is of the preplant treatment main effect (whole plot). For biomass of small fruit and biomass of deformed fruit, only the main effect of treatment was significant, resulting in the data being combined across the two growing seasons.

[f] P-value presented is the main effect of growing season.

[g] P-value represents the interaction between the preplant treatment

of novel soil microbial diversity, function, and metabolic activity [22,59,60]. These microorganisms may boost resistance to fruit damage caused by insects or birds by managing hormones, balancing nutrition, creating plant growth regulators, and activating defense mechanisms [61]. The application of post-plant treatments, including *B. velezensis* IALR619, Terra-Grow, oxidate 0.5 + TerraGrow, or uninoculated, did not demonstrate a statistically significant effect on fruit yield parameters across either season. These results were consistent with our previous research conducted at the Hampton Roads AREC, where no significant differences were observed between bacterial treatments and the control [29]. The application of *Bacillus* spp. as post-plant treatments did not lead to a significant increase in strawberry fruit yield parameters, as these improvements are typically linked to the effective control of fungal diseases [62]. The reduced levels of biotic and abiotic stress observed during both growing seasons may have minimized the overall efficacy of the post-plant treatments. Additional replicated studies at different geographic locations with varied climates and soil types will be needed to better understand these treatments.

### 3.5  Postharvest fruit quality

Firmness, total soluble solids (TSS), and pH are significant factors influencing consumer preference. The interaction between preplant treatments, post-plant treatments, and growing season was not statistically significant for fruit firmness and TSS, and thus, only the main effect data are reported. In contrast, a significant interaction was detected between preplant treatments and post-plant treatments (P = 0.0067), as well as between preplant treatments and growing season for fruit pH (P = 0.0012), necessitating the separate presentation of pH data for each growing season. Regarding fruit firmness, the combination of ASD and *B. velezensis* IALR619 produced slightly firmer fruit (0.10 kg) compared with the non-fumigated treatments (0.08 kg) and the Pic-Clor 60 treatments (0.84 kg). In addition, inoculation with *B. velezensis* IALR619 (0.09 kg) in the non-fumigated control plots significantly enhanced fruit firmness compared to the uninoculated control (Fig 5). Plant growth-promoting bacteria, like *B. velezensis* IALR619, enhance plant growth by secreting hormones such as indole acetic acid, auxin, which promotes root hair and lateral root development, thereby improving nutrient uptake and enhancing fruit quality [63,64]. In addition, *B. velezensis* has been shown to possess phosphate-solubilizing capabilities, which can increase phosphorus availability in the soil, thereby improving root architecture and nutrient uptake, potentially contributing to enhanced TSS in strawberry fruit [65].

The preplant treatments had a significant influence on the TSS (P = <0.0001) and pH of the fruit juice (P = 0.001), whereas the post-plant treatments did not affect these parameters in either growing season. Although there was no significant interaction between the growing season and the preplant treatments for TSS, a significant interaction was found for the pH of the fruit juice data. Consequently, the combined main effect of the preplant treatments is reported across both growing seasons for TSS, whereas the effects of the preplant treatments are presented separately for each season regarding pH data (Figs 6 and 7). The results indicated that the ASD application moderately improved strawberry fruit quality, as reflected by slightly higher TSS and pH values under this treatment. In comparison, the application of the Pic-Clor 60 treatment resulted in the lowest TSS and pH values in both growing seasons. The TSS and pH values for all treatments of the tested strawberries were found to be within the expected range, with TSS measuring between 7.7 and 8.5 °brix and pH ranging from 3.33 to 3.55. These values align with the literature, which reports TSS levels of 4.8 to 10.9 °brix and pH values from 3.20 to 5.40 for strawberries at various ripening stages [66,67].

## 4.  Conclusions

Although chemical pesticides and fumigants are effective for pest control, their high cost, potential for pathogen resistance, residue concerns, emergence of new pathogens, and risks to human and environmental health necessitate alternative strategies. Soil fumigation is particularly challenging for organic and small-scale growers, highlighting the need for biological and biorational approaches. In our two-season field trial, anaerobic soil disinfestation (ASD) consistently reduced AFR, BFR, and weed pressure. The low-cost, readily available carbon source used for ASD

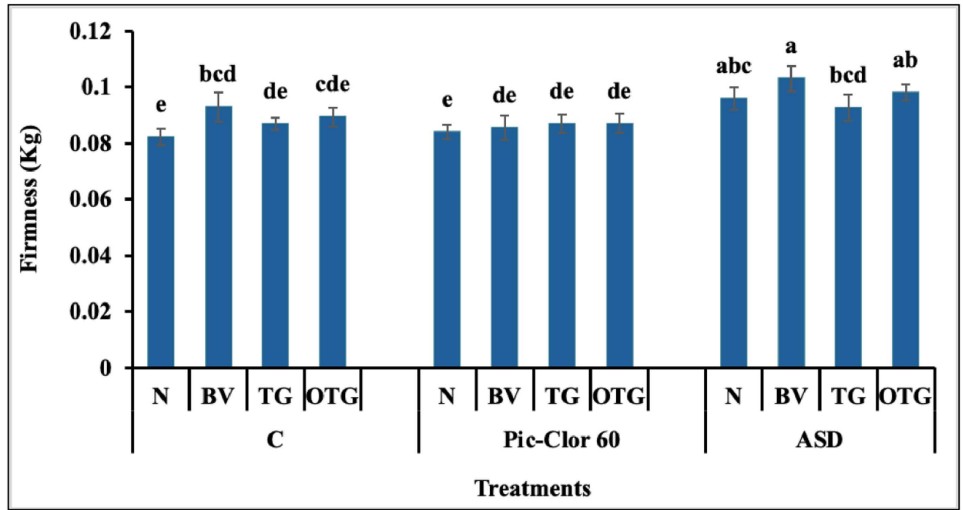

**Fig 5. Effect of preplant treatments and post-plant treatments on fruit firmness.** UTC: non-fumigated control; Pic-Clor 60: [1,3-dichloropropene plus chloropicrin (40:60, w/w)] shank fumigated at 196 kg ha$^{-1}$; ASD: anaerobic soil disinfestation using brewer's spent grain (4.5 tons/ ha$^{-1}$) as nitrogen source and paper mulch (4.8 tons/ ha$^{-1}$) as a carbon source. The post-plant treatments: N: uninoculated control; BV: inoculated with *B. velezensis* IALR619, OD$_{600}$ = 1.0; TG: inoculated with TerraGrow; OTG: inculcated with Oxidate 5.0 + TerraGrow. Means are represented by bars with the same letters, indicating no statistically significant differences ($P < 0.05$) using Fisher's Least Significant Difference (LSD). Error bars indicate the standard error of the mean.

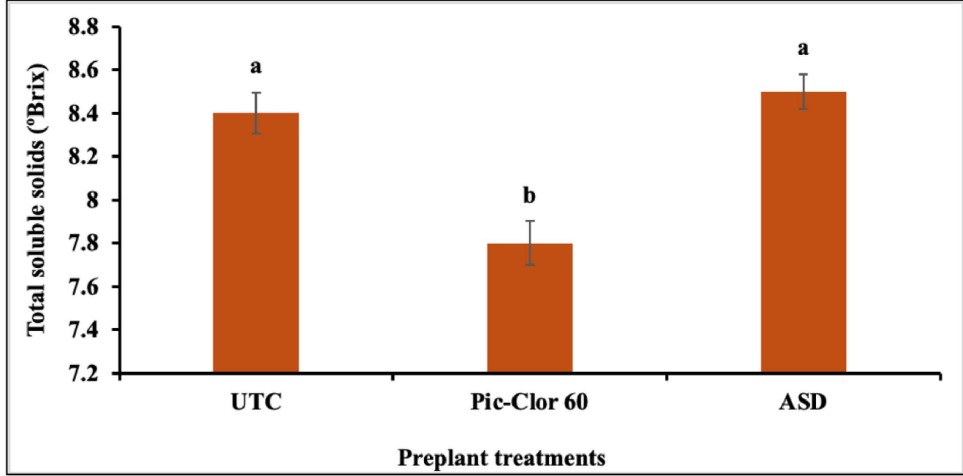

**Fig 6. Effect of preplant treatments on the total soluble solids (TSS) during the 2022/23 and 2023/24 growing seasons.** UTC: non-fumigated control; Pic-Clor 60: [1,3-dichloropropene plus chloropicrin (40:60, w/w)] shank fumigated at 196 kg ha$^{-1}$; ASD: anaerobic soil disinfestation using brewer's spent grain (4.5 tons/ ha$^{-1}$) as nitrogen source and paper mulch (4.8 tons/ ha$^{-1}$) as a carbon source. Means are represented by bars with the same letters, indicating no statistically significant differences ($P < 0.05$) using Fisher's Least Significant Difference (LSD). Error bars indicate the standard error of the mean.

enhances its practical adoption, particularly for resource-limited operations. Combining ASD with beneficial microbes further improved disease suppression and fruit quality, offering a promising strategy for integrated pest management. This approach may be especially valuable in organic systems, small farms, and buffer zones before transplanting. Future research should refine ASD protocol, such as optimizing carbon source rates and incorporating preplant

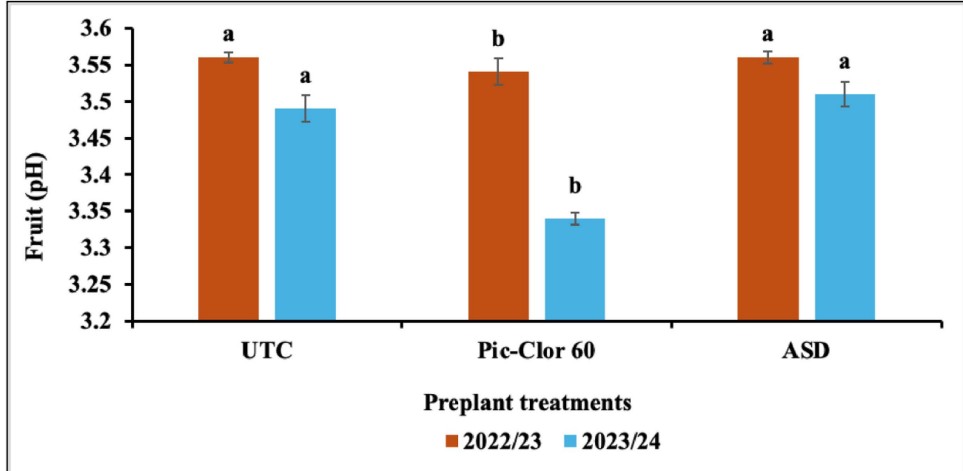

**Fig 7. Effect of preplant treatments on the pH of the fruit juice during the 2022/23 and 2023/24 growing seasons.** UTC: non-fumigated control; Pic-Clor 60: [1,3-dichloropropene plus chloropicrin (40:60, w/w)] shank fumigated at 196 kg ha⁻¹; ASD: anaerobic soil disinfestation using brewer's spent grain (4.5 tons/ ha⁻¹) as nitrogen source and paper mulch (4.8 tons/ ha⁻¹) as a carbon source. Means are represented by bars with the same letters, indicating no statistically significant differences (P < 0.05) using Fisher's Least Significant Difference (LSD). Error bars indicate the standard error of the mean.

fertilizers to enhance yield, and elucidate the mechanisms underlying pathogen suppression for targeted, site-specific recommendations.

## Supporting information

**S1 Fig. Temperatures recorded under ASD and UTC plots during treatement period in 2022/23 and 2023/24 growing season.** UTC = Untreated; ASD = Anaerobic soil disinfestation, Air = Air temperature.
(DOCX)

## Acknowledgments

The authors would like to express their gratitude to Robert Holtz, Jacques Pouliquen, Gabriel Yeboah, Abigail Craige, Audreana Ellis, Amy Burnett, Mia Perry, and Roy Flanagan III for their assistance in conducting the field trial.

## Author contributions

**Conceptualization:** Jayesh B. Samtani.

**Data curation:** Baker D. Aljawasim, Jayesh B. Samtani.

**Formal analysis:** Baker D. Aljawasim.

**Funding acquisition:** Baker D. Aljawasim, Jayesh B. Samtani.

**Investigation:** Baker D. Aljawasim, Patricia Richardson, Chuansheng Mei, Jayesh B. Samtani.

**Methodology:** Baker D. Aljawasim, Robert L. Chretien, Jayesh B. Samtani.

**Project administration:** Jayesh B. Samtani.

**Resources:** Chuansheng Mei, Robert L. Chretien, J. Scott Lowman, Jayesh B. Samtani.

**Software:** Jayesh B. Samtani.

**Supervision:** Patricia Richardson, J. Scott Lowman, Jayesh B. Samtani.

**Validation:** Baker D. Aljawasim, Jayesh B. Samtani.

**Visualization:** Jayesh B. Samtani.

**Writing – original draft:** Baker D. Aljawasim.

**Writing – review & editing:** Baker D. Aljawasim, Patricia Richardson, Chuansheng Mei, Robert L. Chretien, J. Scott Lowman, Jayesh B. Samtani.

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
