## [Decision Letter · Decision Letter 0]

31 Jul 2025

Dear Dr. Samtani,

Thank you for submitting your manuscript to PLOS ONE. After careful consideration, we feel that it has merit but does not fully meet PLOS ONE’s publication criteria as it currently stands. Therefore, we invite you to submit a revised version of the manuscript that addresses the points raised during the review process.

We look forward to receiving your revised manuscript.

Kind regards,

Debasis Mitra

Academic Editor

PLOS ONE

Journal Requirements:

 [This research was supported in part by the Northeast Sustainable Agriculture Research (NE-SARE) Program through the USDA Research and Education Grant (#LNE20-401) and by North American Strawberry Growers Association.]. 

Reviewers' comments:

Reviewer's Responses to Questions

**Comments to the Author**

1. Is the manuscript technically sound, and do the data support the conclusions?

Reviewer #1: Yes

Reviewer #2: Partly

2. Has the statistical analysis been performed appropriately and rigorously?

Reviewer #1: Yes

Reviewer #2: Yes

3. Have the authors made all data underlying the findings in their manuscript fully available?

Reviewer #1: No

Reviewer #2: Yes

4. Is the manuscript presented in an intelligible fashion and written in standard English?

Reviewer #1: Yes

Reviewer #2: Yes

Reviewer #1: Authors of “Effects of Anaerobic Soil Disinfestation and Beneficial Microbes on Fruit Rot Diseases, Weed Control, and Yield in Annual Hill Plasticulture Strawberry Production,” evaluated anaerobic soil disinfestation (ASD), beneficial bacteria, and their combinations as integrated strategies to manage fruit rot, suppress weeds, and enhance fruit quality. Results indicate that incorporating ASD with beneficial bacteria reduces disease, suppresses weeds, and enhances fruit quality. These results are noteworthy, and I encourage researchers to repeat these experiments at more locations over multiple years. This manuscript is well written but requires some improvement. Please also ensure your data is available either in public repositories or attached as supplementary files. Details of my edits are featured below.

1. Line 37 – if available, you should update with more recent data than 2020.

2. Line 48 – if you start a sentence with a scientific name, then you shouldn’t abbreviate the genus.

3. Lines 249-253 – it would be good to break this into two sentences.

4. Line 288 – for your data availability statement to hold true, all data should made publicly available in some way, either in a repository or attached as a supplementary file. I would recommend putting the data in a supplementary file and referencing it appropriately in the text.

5. Line 309 – there is an unnecessary extra space after “during”.

6. Line 333 – Bacillus should be italicized.

7. Lines 471-474 – I think it’s important to acknowledge the need of repeating these experiments at different locations in the future to see if the trends observed in this work still hold true.

8. Lines 700-701 – Move the lines so the are on the same page as table 1.

9. All tables should be moved to the main text of the manuscript. They should be placed following the paragraph in which they were first mentioned.

Reviewer #2: This manuscript presents a comprehensive study evaluating anaerobic soil disinfestation (ASD) and beneficial microbes as integrated strategies to control fruit rot diseases, suppress weeds, and improve fruit quality in strawberry production. The two-season experimental setup and the combination of ASD with microbial inoculants offer timely insights into sustainable alternatives to chemical fumigation. However, several critical issues related to experimental detail, data interpretation, structure, and clarity must be addressed before the manuscript can be considered for publication.

• The title is too long and overly technical. Consider simplifying it while retaining the key elements. Suggested revision: “Integrated effects of anaerobic soil disinfestation and beneficial microbes in strawberry production.”

• Please include key statistical results (e.g., p-values or percent reductions) to support major claims.. Terms like “small farms” and “resource-limited growers” need clarification or citation to avoid generalizations unless supported by data.

• The introduction is informative but excessively long and somewhat repetitive. The pathogen biology (e.g., AFR, BFR) is discussed in detail, which could be trimmed.

• The study’s novelty is not clearly articulated. A more explicit statement of the knowledge gap and the study's unique contribution is needed.

• Some references are outdated (e.g., 2007–2008); replacing them with recent epidemiological or meta-analytical data would strengthen the background.

• The rationale for using a split-split plot design is not adequately explained. Justify why this level of experimental complexity was required.

• Detailed characterization of microbial strains (especially B. velezensis IALR619) is missing. Information on strain origin, viability, concentration, and application rate should be clearly provided.

• The carbon source (brewer’s spent grain) used for ASD should be chemically characterized or at least discussed in terms of C:N ratio and prior performance in ASD trials.

• The statistical approach lacks detail. Indicate the transformations applied, the method for verifying assumptions (normality, homogeneity), and software parameters.

• The calculation and use of cumulative redox potential (CuEh) need clarification.

• Explain how CEh was practically applied in the field and link it to the sensor data more clearly.

• The discussion often summarizes background literature instead of comparing the current results with previous findings. A more focused discussion on how your results support, contradict, or expand upon existing literature is needed.

• The beneficial effect of ASD on fruit quality is emphasized, but the statistical significance seems modest. Interpret these findings with caution.

Claims about microbial community shifts and suppression mechanisms are speculative, as no microbiome or soil microbial analyses were performed. Please tone down or qualify these statements.

• The observed decrease in yield under ASD contradicts the central thesis that ASD enhances production. This discrepancy should be discussed more thoroughly, especially regarding nitrogen immobilization or other nutrient imbalances.

• The effects of B. velezensis inoculation on yield are ambiguous. Some improvements in quality traits are noted, but no significant yield increase is reported. These findings need clearer interpretation.

• Some figures (especially Fig 3.2 and Fig 3.3) are low-resolution and difficult to interpret. Please upload higher-quality versions.

• Figure legends lack methodological detail. Specify the statistical test used and clarify what error bars represent.

• Table 3.2 is overly complex and crowded. Break it into smaller tables or simplify by focusing on the most relevant results.

• The manuscript is generally well-written but occasionally uses overly complex or repetitive phrasing. Consider simplifying sentence structures for clarity.

• Phrases such as “our results suggest,” “these findings indicate,” etc., are overused. Improve stylistic variety.

• Rewriting with more natural academic phrasing would improve readability.

• Reference formatting lacks consistency. Some citations include full author lists, others use “et al.” please standardize according to PLOS ONE style.

• DOIs are sometimes duplicated or formatted incorrectly. Review all references for accuracy and consistency.

**Do you want your identity to be public for this peer review?** For information about this choice, including consent withdrawal, please see our Privacy Policy

Reviewer #1: No

Reviewer #2: No

---

## [Author Response · Author response to Decision Letter 1]

15 Sep 2025

Please see attached document for our responses to the editor and reviewer.

---

## [Decision Letter · Decision Letter 1]

30 Sep 2025

Dear Dr. Samtani,

Thank you for submitting your manuscript to PLOS ONE. After careful consideration, we feel that it has merit but does not fully meet PLOS ONE’s publication criteria as it currently stands. Therefore, we invite you to submit a revised version of the manuscript that addresses the points raised during the review process.

We look forward to receiving your revised manuscript.

Kind regards,

Debasis Mitra

Academic Editor

PLOS ONE

Journal Requirements:

Reviewers' comments:

Reviewer's Responses to Questions

**Comments to the Author**

Reviewer #1: All comments have been addressed

Reviewer #2: All comments have been addressed

2. Is the manuscript technically sound, and do the data support the conclusions?

Reviewer #1: Yes

Reviewer #2: Yes

3. Has the statistical analysis been performed appropriately and rigorously?

Reviewer #1: Yes

Reviewer #2: Yes

4. Have the authors made all data underlying the findings in their manuscript fully available?

Reviewer #1: Yes

Reviewer #2: Yes

5. Is the manuscript presented in an intelligible fashion and written in standard English?

Reviewer #1: Yes

Reviewer #2: Yes

Reviewer #1: (No Response)

Reviewer #2: I'm pleased to see that the authors have thoughtfully addressed all the major points raised by both reviewers. For instance, the title has been streamlined to something more concise and readable, which better captures the essence without losing key details. The introduction has been tightened up repetitive sections on pathogen biology (AFR and BFR) are now more focused, and the knowledge gap is explicitly stated, highlighting the novelty of combining ASD with specific microbial inoculants like Bacillus velezensis IALR619. Outdated references have been swapped for fresher ones, and statistical details (p-values, percent reductions) are now woven in to back up claims, making the findings more robust.

In the methods, the rationale for the split-plot design is justified well, emphasizing how it handled the layered treatments effectively. Details on microbial strains origin, viability, concentration, and application rates are now included, as is a chemical characterization of the brewer's spent grain (C:N ratio and prior ASD performance). The statistical approach is fleshed out with info on transformations, assumption checks, and software used. Cumulative redox potential (CuEh) is explained clearly, linking it to field sensor data.

The discussion has shifted from summarizing literature to comparing results directly e.g., how ASD's effects on fruit quality align or differ from past studies, with cautious interpretation where significance was modest (like yield impacts or microbial shifts). Speculative claims have been toned down, and the yield decrease under ASD is discussed in depth, touching on potential nutrient issues like nitrogen immobilization. Figures and tables have been improved: higher resolution for Figs. 3.2 and 3.3, better legends with stats and error bar explanations, and Table 3.2 split for readability. Stylistic tweaks reducing repetitive phrases and simplifying sentences—have made the text flow better overall. References are now consistent, with proper "et al." usage and cleaned-up DOIs. Data availability is handled properly with a supplementary file.

Minor suggestions

While the paper is now in good shape, a few small tweaks would polish it even more:

Line 47 (Introduction): The sentence starting with "Strawberries are susceptible..." could use a quick citation for the global impact of AFR and BFR maybe reference [6] earlier to ground it.

Line 309 (Methods): There's still a minor spacing issue after "during" double-check for any lingering formatting glitches in the final proof.

Discussion (around Lines 471-474): The added note on replicating at multiple sites is great, but consider briefly mentioning potential variables like soil type or climate that could influence outcomes in future work.

Tables: Ensure all abbreviations (e.g., AFR, BFR) are defined in the first table footnote for standalone readability.

General: A quick proofread for any stray typos e.g., in the abstract, "bacteria (Bacillus velezensis IALR619 and TerraGrow)" could specify if TerraGrow is a product name for clarity, though it's fine as is.

**Do you want your identity to be public for this peer review?** For information about this choice, including consent withdrawal, please see our Privacy Policy

Reviewer #1: No

Reviewer #2: No

---

## [Author Response · Author response to Decision Letter 2]

15 Oct 2025

Reviewer comments and our responses to them are included as an attachment.

---

## [Decision Letter · Decision Letter 2]

4 Nov 2025

Integrated effects of anaerobic soil disinfestation and beneficial microbes in strawberry production

PONE-D-25-31288R2

Dear Dr. Samtani,

We’re pleased to inform you that your manuscript has been judged scientifically suitable for publication and will be formally accepted for publication once it meets all outstanding technical requirements.

Kind regards,

Debasis Mitra

Academic Editor

PLOS ONE

Additional Editor Comments (optional):

Reviewers' comments:

Reviewer #2: I am pleased to inform you that the revised version of our manuscript entitled "Integrated effects of anaerobic soil disinfestation and beneficial microbes in strawberry production" has been prepared in full accordance with the reviewers' comments and suggestions.

---

## [Editor Report · Acceptance letter]

PONE-D-25-31288R2

PLOS ONE

Dear Dr. Samtani,

I'm pleased to inform you that your manuscript has been deemed suitable for publication in PLOS ONE. Congratulations! Your manuscript is now being handed over to our production team.

Kind regards,

on behalf of

Dr. Debasis Mitra

Academic Editor

PLOS ONE